# Deriving First Floor Elevations within Residential Communities Located in Galveston Using UAS Based Data

**Nicholas D. Diaz** [1,*], **Wesley E. Highfield** [1], **Samuel D. Brody** [1] and **Brent R. Fortenberry** [2]

1   The Center for Texas Beaches and Shores (CTBS), Texas A&M University at Galveston, Galveston, TX 77554, USA; highfiew@tamug.edu (W.E.H.); brodys@tamug.edu (S.D.B.)
2   School of Architecture, Tulane University, New Orleans, LA 70118, USA; bfortenberry@tulane.edu
*   Correspondence: nicholas.diaz9@tamu.edu; Tel.: +1-512-422-9491

**Abstract:** Flood damages occur when just one inch of water enters a residential household and models of flood damage estimation are sensitive to first-floor elevation (FFE). The current sources for FFEs consist of costly survey-based elevation certificates (ECs) or assumptions based on year built, foundation type, and flood zone. We sought to address these limitations by establishing the role of an Unmanned Aerial System (UAS) to efficiently derive accurate FFEs. Four residential communities within Galveston Island, Texas were selected to assess efficient flight parameters required for UAS photogrammetry within the built environment. A real-time kinematic positioning enabled (RTK) UAS was then used to gather georeferenced aerial imagery and create detailed 3D photogrammetric models with ±0.02 m horizontal and ±0.05 m vertical accuracies. From these residential models, FFEs and other structural measurements present in traditional ECs were obtained. Comparative statistical analyses were performed using the UAS-based measurements and traditional EC measurements. UAS based FFE measurements achieved 0.16 m mean absolute error (MAE) across all comparative observations and were not statistically different from traditional EC measures. We conclude the RTK enabled UAS approach is an efficient, cost-effective method in establishing accurate FFEs and other flood-sensitive measures in residential communities.

**Keywords:** first-floor elevation (FFE); Unmanned Aerial System (UAS); Structure-from-Motion (SfM); photogrammetry; flood risk reduction; UAS flight parameters

## 1. Introduction

### 1.1. Motivation and Data Gaps

Coastal margins are experiencing increased economic and environmental loss due to various anthropogenic and global changes [1–5]. Urban development has increased populations within coastal counties and climate changes (i.e., sea-level rise, storm intensification) have, and will continue to, cause higher flood risks and damages. Coastal and hazard planners rely on spatial and elevational datasets across multiple scales as the basis to fulfill geomatic practices (GIS-based mapping), scenario planning, and mitigation and adaption strategies [1–4]. Available large-scale datasets make accurate measurements and damage assessments at smaller scales difficult, thus providing decision-makers with data gaps for vulnerable communities and high-interest areas. Gathering data at smaller scales can be costly depending on the extent of the area, resolution, and equipment used. For example, the use of an airplane for aerial scanning or imagery for minimal transects of an area could cost upward of 20k USD. Traditional methods for collecting key elevational measures on households are time- and cost-intensive, as seen in Section 1.3. To address the lack of alternatives for a cost-effective, efficient measurement tool in gathering spatial and elevational data, this study proposed the use of an Unmanned Aerial System (UAS) coupled with photogrammetry methods to capture small-scale data with a focus on small-scale elevational measurements critical to assessing flood inundation risk and damages.

### 1.2. UAS Technology for PaRS and 3D Modeling

UASs, often referred to as drones, have undoubtedly created a competitive market niche with various applications [6–9]. UAS assist in gaining missing information or monitoring on-going functions such as Non-Military Governmental (civil security), emergency response services (forest fire spotting), energy and communication networks (pipeline monitoring), agriculture and fisheries (crop and population monitoring), engineering practices (structure integrity), cultural heritage (archeological site reconstruction), and general intelligence, surveillance, and reconnaissance. Regardless of the UAS type, fixed-winged or multirotor, these systems have evolved with technology gaining new functions geared towards geomatic applications such as photogrammetry and remote sensing (PaRS) and 3D modeling [10,11].

Photogrammetry and remote sensing involve various photographic and scanning methods to understand the geospatial and topographic characteristics of an area [10,12]. Photogrammetry often utilizes passive remote sensing techniques that involve capturing aerial imagery with natural light, such as Structure-from-Motion (SfM) imagery. SfM imagery is overlapping, offset images of an area or 3D structure. These images can then be used for photogrammetric processing, which uses the captured 2D images to create 3D models factoring in UAS flight altitude, camera type, and gimbal angle. Most photogrammetric software packages can automatically detect these parameters through data files attached to the photo population. Previous studies have shown SfM methods are cost and labor-friendly relative to active remote sensing techniques, such as airborne Light Detection and Ranging (LiDAR), while still producing high-quality dense point clouds (DPCs) and 3D models [12,13]. If the correct and efficient methods of operating the UAS for SfM imagery are maximized, similar model generation and accuracies would occur compared to costlier LiDAR UAS approaches.

The accuracy of the resulting DPCs and 3D models can vary based on flight parameters and equipment used [7,11,14]. Traditional methods using RTK enabled GPS and ground control points (GCPs) allow 3D models to come within ±2.5 cm horizontal and ±5 cm vertical errors. However, to reach these accuracies, up to 40 GCPs had to be established [15,16]. Additional UAS vertical errors (±0.066 m and ±0.088 m) are reported using 1.5 to 6 GCPs per hectare [17,18]. Attainment of these errors following similar method requirements calls for increased time and labor in the field during deployment and after post-processing within photogrammetry software. Many of these digital elevation models (DEMs) were also environmental and low-relief (2.5D), suggesting that higher relief areas (built environment) would require even more GCPs and post-processing [19].

Other low-relief UAS studies showed ±0.3 m elevational accuracies when using total station theodolite (TST) data [20,21]. Structural recreation modeling within the built environment can be modeled using UAS technology as seen with historical monuments and archeological sites [14,22,23]. Accurate 3D DPCs and models were created by combining methods of UAS SfM capture and terrestrial laser scanning (TLS) to establish 117 GCPs for just one structure. Aside from deformation and construction monitoring [11,24], UAS technology has not been utilized to capture flood-sensitive measures within the built environment. This study proposes the use of an RTK enabled UAS (see Section 2.3), which dramatically reduces the number of GCPs required to accurately model DPCs, ultimately reducing time and labor.

### 1.3. First Floor Elevations (FFEs) and FEMA-NFIP Guidelines

The Federal Emergency Management Agency (FEMA) recognizes the lowest floor elevation (LFE), or first-floor elevation (FFE), as the lowest enclosed area (including basement) [25]. However, depending on the structure, architectural style, and foundation type, the lowest floor can be hard to determine and interpret, even when using the National Flood Insurance Policy (NFIP) guidance [25,26]. Under the NFIP, all new and substantially improved structures must have the lowest floor elevated to or above the Base Flood Elevation (BFE). This is monitored by utilizing private survey contractors to measure and sign

elevation certificates (ECs). Non-residential buildings may be floodproofed below the BFE. The BFE, along with flood zones, are shown on FEMA's Digital Flood Insurance Rate Maps (FIRM) and are used to establish flood insurance premiums. FEMA-based damage curves also recognize that just 1 inch of water above the BFE costs on average $26,000 in flood damages [25,26].

The two primary flood zones within coastal margins are A and V zones. A-zones are high-risk areas subject to water level rise or inundation by the 1-percent-annual-chance flood event [27]. V-zones are described as high-risk coastal areas subject to inundation by the 1-percent-annual-chance flood event with additional hazards associated with storm-induced waves. Both zones result in mandatory flood insurance purchase requirements and floodplain management standards. ECs allow residential and non-residential structures to comply with these requirements and provide insurers the necessary information to calculate insurance rates in relation to the BFE zones. The ECs contain elevation information including the lowest floor, or FFE (C2.a), the second floor if applicable (C2.b), the lowest horizontal member (C2.c), any grade elevation supporting utilities or service to the house (LSG) (C2.e), the lowest adjacent grade (LAG) (C2.f), and the highest adjacent grade (HAG) (C2.g) (Figure 1) [27].

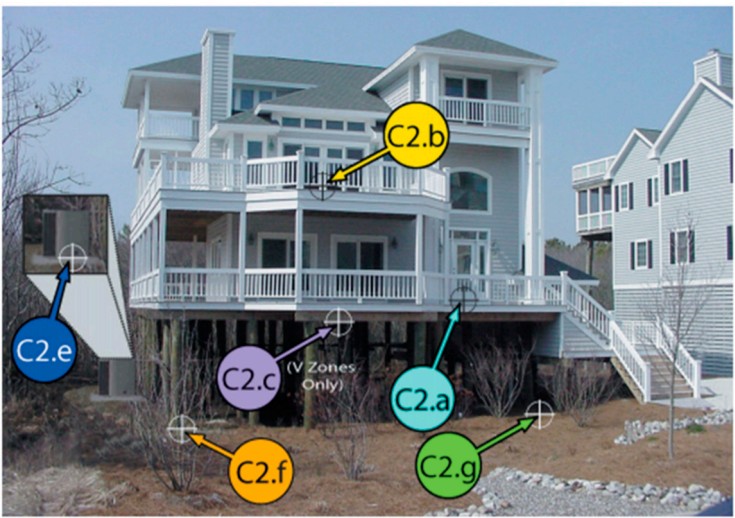

**Figure 1.** EC measures indicated on an example home [25].

A licensed surveyor, engineer, or architect uses a total station theodolite (TST) and must complete, seal, and submit the EC to the community code official [25]. The TST cost range can be 2k–10k USD not including time, labor, and other equipment associated with capturing the EC measures. Not placing the lowest supporting horizontal members and the first floor of a building at the proper elevation in a coastal area can be extremely costly and difficult to correct. Note that multiple ECs may need to be submitted for the same building: a certificate may be required when the lowest floor level is set (and before additional vertical construction is carried out); a final certificate must be submitted upon completion of all construction prior to issuance of the certificate of occupancy (COs) [25].

Aside from flood insurance requirements, individual inundation risks are also based on where water will enter a home. FFE estimates are primary measurements that explain where and when individuals can expect potential harm from flood events. Spatial flood risk modeling is challenging and is often limited by available data or the quality of data acquired [28–30]. Extensive data management across multiple fields (i.e., hydrological, hydraulic, historical, geological, etc.) is required to merge and feed into scenario-based and probabilistic flood assessments.

Hybrid approaches to flood modeling have been explored using GIS-based and neuro-fuzzy methods to decrease uncertainty [31–33]. Other regional hybrid approaches involve the use of high-resolution ortho-imagery, LiDAR, and Google Street View (GSV) imagery

to capture parcel classifications and measurements [34]. FFEs can provide a new available quantitative data field able to be coupled with both regional and parcel hybrid flood risk modeling approaches to identify and quantify flood risk at the individual level more accurately. Additional flood-sensitive fields can also be obtained using the UAS-based approach, such as second floor (if any) and LSG. The augmentation of already existing approaches using flood-sensitive data such as FFEs, second floors, and LSGs could further assess the individual risk not only in current time but also matched with future storm intensification and sea-level rise curves [5,35,36].

### 1.4. Research Questions

ECs within Galveston are a function of house modifications through COs or private insurance contracts. Therefore, easily accessible public FFE data of housing units on Galveston Island are limited, a case that is common in the US. This research aimed to test if the use of an RTK-enabled UAS coupled with photogrammetric methods was viable in determining FFEs, and other flood-sensitive measures, of comparable accuracy to traditional EC surveying. The viability of a UAS method would mean obtaining FFEs would no longer be limited to individual household survey-based ECs under the NFIP, and enable large-scale community-based FFE acquisition. To address the lack of alternative measurement methods for obtaining FFEs, estimation of FFEs was sought using an UAS-based approach. More specifically, we address the following research question: Can the use of a UAS coupled with photogrammetry methods survey FFEs and other flood-sensitive measures with accuracy comparable to traditional EC methods?

This research did not aim to test if the UAS was accurate. Rather, it aims to test if the use of the UAS was accurate in capturing the same measures that are recorded in ECs, ultimately to use this information for broader planning purposes aside from individual homeowner's flood insurance policies.

## 2. Data Acquisition and Methods

### 2.1. EC and UAS Data

This study utilized a UAS to capture SfM imagery and create accurate 3D models of selected residential communities in Galveston. Higher-resolution aerial imagery and photogrammetric modeling were necessary for obtaining accurate, 'hard-to-get' measurements related to inundation risk and damages, such as FFEs of households. COs and ECs were provided by the City of Galveston planning and Development Division. The city keeps ECs on file when a home has been newly constructed or when the property owner is seeking permits to raise the structure out of the flood plain. Electronic copies of data available date back to January of 2012.

A dataset of 70 ECs was compiled by cross-referencing COs given by the City of Galveston and addresses located in the study areas chosen. The data provided within these ECs along with data collected using the UAS were the two sources used in this study. Specifically, the five most prominent measurements recorded in ECs including (1) top of the bottom floor (FFE or C2.a), (2) top of the next highest floor (C2.b), (3) lowest service grade (LSG or C2.e), (4) lowest adjacent grade (LAG or C2.f), and (5) highest adjacent grade (HAG or C2.g) (see Figure 1 above) were used [25].

### 2.2. Site Selection

Galveston Island offers diverse residential communities due to the plethora of historical development and natural disasters. Coastal Texas, and especially Galveston Island, have experienced a large share of hurricanes that have developed in the Atlantic and Gulf of Mexico [37]. Major landfalling hurricanes specific to Galveston include Alicia (1983) and Ike (2008), causing approximately 3 billion USD and 27 billion USD in damages, respectively. Hurricane Harvey (2017), also caused 125 billion USD in damages, was primarily a function of rainfall duration, rather than incursions of storm surge [38,39]. Many houses remain slab on grade while others are raised or modified to mitigate against future storms. Slab on

grade foundation households exhibit concrete, non-elevated, or raised foundations. Raised and pier foundations households exhibit floors elevated above the ground by wood stilts or masonry supports.

Four residential communities located on Galveston Island were selected as study sites: (1) Lafitte's Cove, (2) Campeche Cove, (3) Evia, and (4) Silk Stocking District (Figure 2). Each community contained approximately 200–300 housing units providing diversity in multiple dimensions, including location on the island, foundation types, and year built (Table 1). These dimensions were important in assessing the UAS approach with regards to efficiency and measurement obtainment in large-scale, diverse residential community settings.

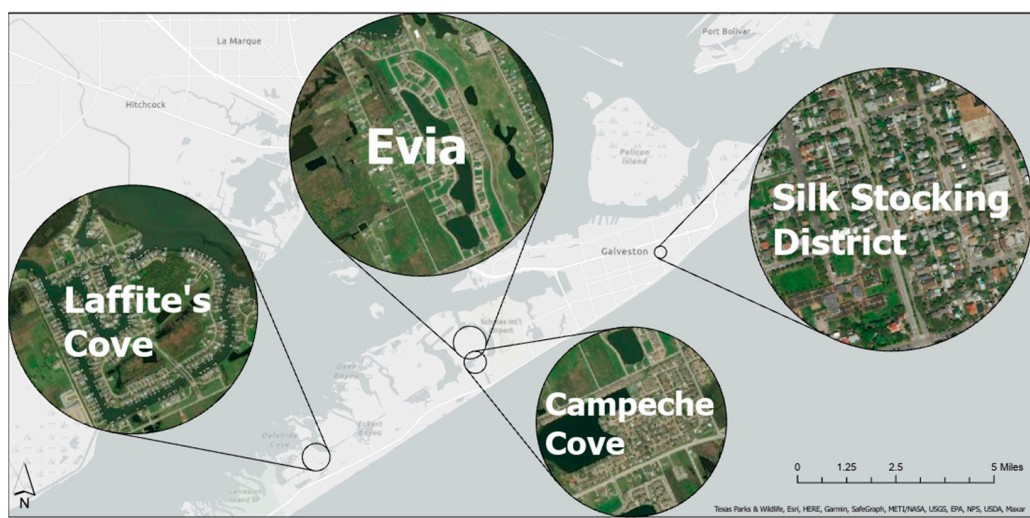

**Figure 2.** Map of selected residential communities located in Galveston, TX.

**Table 1.** Selected residential community housing information.

| Community | Foundation Types | Development Years |
|---|---|---|
| Lafitte's Cove | Pier | 1996–Current |
| Campeche Cove | Slab, Raised | 1978–2004 |
| Evia | Slab, Raised, Pier | 2004–Current |
| Silk-Stocking District | Slab, Raised, Pier | 1890–1975 |

These study sites represent where the UAS SfM flight parameter testing occurred and where the UAS was used to perform aerial surveys for the photogrammetric workflows. The photogrammetric survey imagery capture was performed in each of the four study areas to obtain UAS based EC measurements to compare to the traditional EC measurements provided by the City of Galveston.

*2.3. UAS Flight Parameter Testing and Data Collection*

Remote data collection was performed using a DJI Phantom 4 RTK UAS. As seen in previous studies [15,16], UAS flight parameters are tested to assess the accuracy and quality of 3D models prior to fulfilling the overall study. Drone flight parameters affect many other functions and results including flight time, data storage, photogrammetry software processing, and DPC generation. Given the DJI Phantom 4 RTK was a new product at the time, testing and confirmation of flight parameters were necessary to achieve high accuracies within Pix4D products, especially with regards to large-scale, densely built settings in which seldom studies operate within. Eight UAS flight parameters were tested (see Table 2). Testing these parameters strived for efficient use of the UAS, DPCs with ±5 cm vertical accuracies or better, and DPCs with the necessary points for the derivation of flood-sensitive measures.

**Table 2.** Flight parameters tested using the DJI Phantom 4 RTK.

| Parameter | Description |
|---|---|
| Flight plan type | 3D photogrammetry flight plans including 'double-grid 'and 'multi-oriented' |
| Flight altitude | Altitude (m) at which the UAS captures aerial imagery |
| Flight plan area | area (length and width) of flight plan type |
| Flight plan margin | flight plan boundaries the UAS can or cannot exceed from flight plan area |
| Gimbal tilt | tilt of the gimbal positioning the camera capturing SfM imagery |
| Lateral and front overlap | 2D/Nadir (90-degree gimbal tilt) photo capture overlap percentage |
| Oblique lateral and front overlap | 3D (45–60-degree gimbal tilt) photo capture overlap percentage |
| Start and end points | Start and end locations of the flight plan relative to launch point |

The following explains the key findings regarding flight parameters utilizing the UAS in the selected sites. (1) The flight plan type 'double-grid' was the most efficient in capturing a large number of households within a large area given battery limitations. 'Multi-oriented' flight plan type was the most effective in capturing imagery to model one structure in detail. 'Multi-oriented' was time and battery intensive and did not adequately capture the inner portions of flight plans that covered a large area with multiple households. (2) Flight altitudes were tested from the UAS operator Federal Aviation Administration (FAA) maximum of 120 m to 25 m, which is the device minimum. Operating at altitudes greater than 100 m did not allow for image capture that could accurately or adequately model structures and the information sought on structures (i.e., first floor elevations) either due to roof or tree coverage. It was determined that operating at 45–60 m flight altitude range was necessary for densely built communities, such as the Silk-Stocking District (~10 parcels/acre). Operating lower than 45 m was time and battery intensive as well as increased safety and equipment risks due to obstructions (i.e., telephone lines, trees, commercial buildings). Operating at flight altitudes of 60–100 m was sufficient for less clustered or densely built communities such as Lafitte's Cove (~5 parcels/acre). (3) Although the flight plan area is a function of the study area, it was determined that multiple flight plans could be created to complete a larger area. This was primarily due to battery limitations. If one flight plan could not be completed, it was best to section off smaller areas. (4) Flight plan margin was best left as 'AUTO'. This setting could be slightly changed to save battery or time; however, this would potentially sacrifice exterior oblique capture of the area. (5) Gimbal tilt was best left at 'AUTO', or 60 degrees. If operating below 60 m flight altitude, lowering gimbal tilt was effective in increasing oblique capture. A 1:1 (altitude (m):gimbal tilt (degrees)) 'rule-of-thumb' was determined for this relationship (i.e., 45 degrees at 45 m as the minimum). (6) The use of nadir lateral (%) and front overlap (%) was unnecessary for most residential areas. It was determined useful in densely built, clustered areas such as in the Silk-Stocking District. When used, 70%/70% overlapping rates or greater were determined sufficient. (7) Oblique lateral (%) and front (%) image overlaps using the 'double grid' flight plan were left at the default settings of 80%/80% overlapping rates, however, 70%/70% or greater could be utilized to increase efficiency. Overlap percentages less than 70%/70% did not capture enough photos to generate key points for measurement derivation within the DPCs. Operating at greater than 80%/80% overlaps dramatically increased flight and processing times, as well as data storage. (8) Start and end points of the flight plan did not affect DPC generation; however, they did have the ability to affect flight time and battery efficiency. It was found that start and end points either closest to launch or central to the full area were most efficient. Table 3 shows flight parameters used for each residential community selected in this study.

**Table 3.** UAS flight parameters for each residential community. *: nadir 70/70 percent overlap combined with oblique capture.

| Residential Community | Flight Plan Type | Flight Plan Altitude (m) | Flight Plan Area | Flight Plan Margin | Gimbal Tilt (Degrees) | Oblique Overlap (%/%) | Start and End Points |
|---|---|---|---|---|---|---|---|
| Laffite's Cove | Double Grid | 100 | 4 sections (~40–50 acres each) | AUTO | 60 | 70/75 | Closest to launch location. Center of full 4 sections. |
| Campeche Cove | Double Grid | 100 | 1 section (~61 acres) | 10 m | 60 | 70/80 | Closest to launch location. |
| Evia | Double Grid | 100 | 2 sections (~60 & 115 acres) | AUTO | 60 | 70/80 | Closest to launch locations center of each section |
| Silk-Stocking District | Double Grid | 45 | 4 sections (~10 acres each) | AUTO | 45 | * 80/80 | Closest to launch location. Center of 4 sections. |

Imagery captured by the UAS after flight plan completion was saved and backed up to ensure data security. Approximately half a terabyte worth of storage was used to save and back-up data. A total of 7942 photos were captured across the four selected sites. Note the image count increased as flight altitude and gimbal tilt decreased (Table 4). The ECs provided by the City of Galveston were converted to match the elevational units captured by the UAS. Using NOAA's vertical datum conversion tool, the Galveston EC elevations were validated and converted from NAVD88 (North American Vertical Datum 1988) survey feet to WGS84 (World Geodetic System 1984) meters [40]. Other data stored included RTK geolocation information and Pix4D post-processed files.

**Table 4.** Image counts for each residential community.

| Residential Community | Average Image Count per Section | Total Image Count | Total Acres | Batteries Used | Flight Altitude | Gimbal Tilt |
|---|---|---|---|---|---|---|
| Laffite's Cove | 391 (4 sections) | 1564 | 190 acres | 8 | 100 | 60 |
| Campeche Cove | 944 | 944 | 61 acres | 3 | 100 | 60 |
| Evia | 1158 (2 sections) | 2315 | 182 acres | 9 | 100 | 60 |
| Silk-Stocking District | 780 (4 sections) | 3119 | 40 acres | 8 | 45 | 45 |

*2.4. Data Calibration and Processing*

Once image capture and data security were completed, the images were uploaded to the photogrammetry software Pix4D. Images were then calibrated, geo-oriented, and photogrammetrically stitched together to create a 3D model of the residential communities. To ensure accuracy, post-processing kinematic (PPK) efforts such as including five 3D GCPs were added and performed. Once models were re-optimized, FFEs and other EC measurements of each individual housing unit were then manually collected within the DPC. The X, Y, and Z position of the points within the DPC of the Pix4D 3D model was converted to northings, eastings, and meters above MSL using the UAS networking files.

Interpretations of FFEs and other measure estimates captured using UAS based data and Pix4D models were limited to exterior SfM capture. Additionally, through the lens of individual flood risk, it was assumed that homeowners would not live or gather in enclosed floors such as garages, stairways, or storage areas during a flood event. However, it is acknowledged that these areas are subject to flood damages under the NFIP. Figure 3 shows the interpretations of data points collected and how measures were derived within Pix4D. Interpretations of the FFE estimates were derived using points at the bottom of front, side, and back doors, often balconies and patios connected to these points were assigned the same value as the bottom of doors. Attached garages or enclosures below elevated floors were not used as FFE. If present, the next highest, or second floor, was also determined

using points at the bottom of doors or balconies. LSG measures used points at the top of the grade supporting service units such as A/C and heater units. LAG values were points located at the end of driveways or stairs connecting to the public road or sidewalk. Lastly, HAG values were points at the top of the driveway directly adjacent to the home or garage.

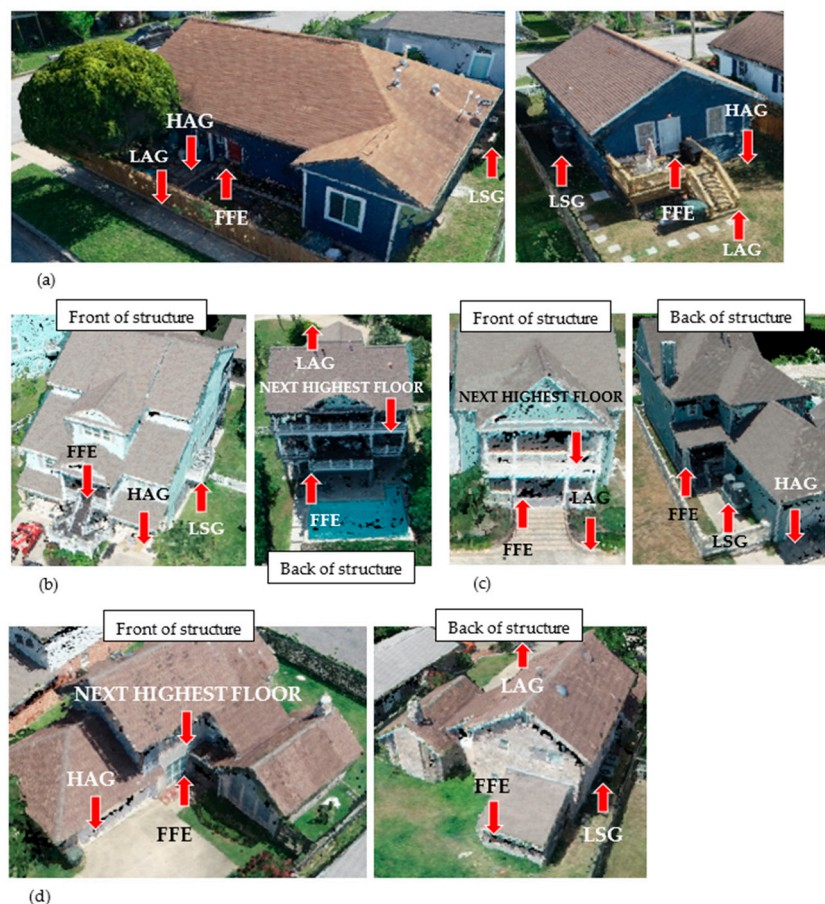

**Figure 3.** Interpretations of measures derived from the residential community models produced within Pix4D. Example shown in tiles (**a**) Silk-stocking district, (**b**) Laffite's Cove, (**c**) Evia, (**d**) Campeche Cove.

Validation of the accuracy in measures was performed by selecting multiple points of a particular measure within Pix4D. The majority of observations had points of equal values across multiple measured locations (i.e., front, side, and backdoor). If the multiple points had a difference in value within 0.05 m, the recorded measure was the average of multiple values. If the multiple points had a difference in value greater than 0.05 m, which rarely occurred, the lower of the two values was recorded. The use of 0.05 m as an accuracy threshold was due to the accuracy of the UAS and subsequent SfM models.

*2.5. Comparative Analysis*

Measurements derived from Pix4D models were paired with their respective EC data. The two datasets across the five EC measures (FFE, next highest floor, LSG, LAG, and HAG) were compared using mean absolute error (MAE) and paired t-tests to determine if the measures were statistically different.

**3. Results**

*Comparative Analysis*

Four ECs were excluded from the original sample of 70 ECs provided due to repeated addresses or lack of sufficient data measures, reducing the sample size to 66 total ECs. Here,

47 ECs recorded addresses located in the Evia Community while the remaining 19 recorded addresses in Laffite's Cove. UAS based 'Top of bottom floor (FFE)' measurements achieved 0.16 m mean absolute error (MAE) and were not statistically different from EC measures (Table 5). UAS based 'Top of next highest floor' measurements achieved 0.31 m MAE and were statistically different from EC measures.

**Table 5.** Mean absolute error (MAE) and paired t-test results achieved using UAS based measures compared with traditional EC measurements.

|  | **Top of Bottom Floor (FFE)** | **Top of Next Highest Floor** | **Lowest Service Grade (LSG)** | **Lowest Adjacent Grade (LAG)** | **Highest Adjacent Grade (HAG)** |
|---|---|---|---|---|---|
| MAE | 0.16 m | 0.31 m | 0.44 m | 0.50 m | 0.19 m |
| *p*-value | 0.832 | 0.034 | 0.768 | 0.000 | 0.143 |

UAS based 'Lowest service grade (LSG)' measurements achieved 0.44 m MAE and were not statistically different from EC measures. UAS based 'Lowest adjacent grade (LAG)' measurements achieved 0.50 m MAE and were statistically different from EC measures. UAS based 'Highest adjacent grade (HAG)' measurements achieved 0.19 m MAE and were not statistically different from EC measures. Statistically significant different 'Next highest floor' and 'LAG' measures could have been a function of measure location and interpretation differences. For example, EC surveyors typically capture measurements on one face of the house and could establish the TST on either the back or front side of the house. If a housing structure differed from front to back, measures were only captured on the side the TST was established. Interpretation of 'Next highest floor' and 'LAG' measures can also be somewhat ambiguous as they are not easily identifiable compared to 'LSG' and 'HAG' measures. Further explanation of measure location and interpretation differences can be seen in the Discussion section.

## 4. Discussion

The use of a UAS for SfM imagery and photogrammetry produced accurate 3D models of the selected residential communities, however, some challenges were discovered during data derivation. Time to derive measurements within these models ranged 2–5 min per household, taking up much of the labor time involved with deriving the UAS based data. The UAS based approach was able to capture four residential communities (952 households) in just under 18 working business days (Table 6).

**Table 6.** Field, processing, and data derivation labor times for each residential community.

| Community | **Field Labor Time** | **Pix4D Processing Time** | **Data Derivation Time** | **Houses Captured** |
|---|---|---|---|---|
| Laffite's Cove | 6.5 h | 10 h | 23.3 h | 280 |
| Campeche Cove | 2.5 h | 3 h total | 24.4 h | 293 |
| Evia | 8 h | 8 h | 17 h | 205 |
| Silk-Stocking District | 5 h | 20 h | 14.5 h | 174 |
| Total | 22 h | 41 h | 79.2 h | 952 |

Interpretation differences and inconsistencies in EC measures also presented a challenge. Initial comparison of measures resulted in 33 height-wise error outliers. Of these, 15 outliers were located in the Evia community while the remaining 18 outliers were located in Laffite's Cove. After further investigation and confirmation, it was discovered that EC measurements were not consistent with other EC measure interpretations, as well as UAS-based measure interpretations limited to exterior SfM capture within FIRM A and V zones. This did not mean that the outlying measures were wrong in their EC evaluation, rather an

observation in the inconsistencies between other surveyors and RTK-based interpretations of the different measure types. For example, see Table 7 and Figure 4 below.

**Table 7.** Interpretation inconsistency example responsible for height-wise errors within the Evia Community. Bold values represent corresponding measures due to interpretation inconsistencies utilizing the two data sources.

| Address | EC FFE [1] (m) | UAS FFE [2] (m) | Error [(1−2)] (m) | UAS HAG (m) |
|---|---|---|---|---|
| 13 Sunrise Row | **3.32** | 4.25 | 0.93 | **3.32** |
| 11 Sunrise Row | 4.21 | 4.28 | 0.07 | 3.36 |
| 9 Sunrise Row | 4.21 | 4.22 | 0.01 | 3.45 |

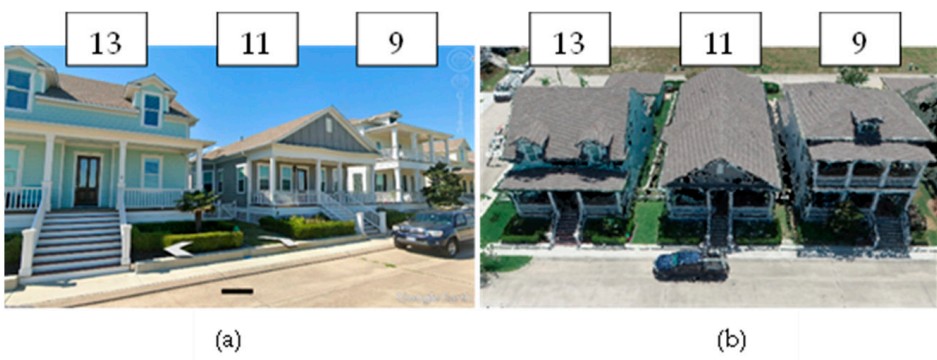

**Figure 4.** Housing addresses Sunrise Row 13, 11, and 9 located in the Evia community seen in (**a**) google street view (GSV) and (**b**) screenshot of Pix4D DPC.

The FFEs for housing addresses Sunrise Row 9, 11, and 13 are all approximately the same elevation. However, 13 Sunrise Row EC FFE was measured dramatically lower than the other two adjacent addresses resulting in a higher height-wise error when compared to the UAS measure. Given that the EC FFE of 13 Sunrise Row exactly matches the elevation of the UAS HAG, it was determined that the EC was most likely captured from the backside of the house. According to NFIP lowest floor guidelines within A zones, attached garages to raised or elevated floors are considered the lowest floor. Essentially, error outliers meant EC FFE measures corresponded to the UAS based HAG measures while the EC next highest floor measures corresponded to the UAS based FFE measures.

The same interpretation inconsistencies were identified for the Lafitte's Cove outliers. However, Lafitte's Cove is split between both A and V zones making interpretations of the FFE especially difficult. For example, although many of the lower floors within pier foundation houses located in Laffite's Cove were enclosed, some of these floors are used for car storage or other storage purposes and are not livable, serviceable floors. Therefore, UAS measures did not record these floors as FFEs, rather HAG measures. There are many criteria for what classifies as an enclosed floor under the NFIP lowest floor guide within V zones. Determination and interpretation of this measure are performed in person across different surveyors. Table 8 and Figure 5 below show an example of interpretation inconsistency within Laffite's Cove.

**Table 8.** EC and UAS interpretation inconsistency examples responsible for large height-wise errors within Laffite's Cove. Grey background columns show height-wise error between different measure types indicated by superscripts.

| Address | EC FFE (m) [1] | UAS FFE (m) [2] | Error [(1−2)] (m) | EC Next Highest Floor [3] | Error [(2−3)] (m) | UAS HAG [4] | Error [(1−4)] (m) |
|---|---|---|---|---|---|---|---|
| 3443 Eckert Drive | 1.64 | 5.97 | 4.33 | 5.96 | 0.01 | 1.55 | 0.09 |
| 3439 Eckert Drive | 1.8 | 6.05 | 4.25 | 6.13 | 0.08 | 1.6 | 0.2 |
| 3410 Eckert Drive | 2.22 | 5.88 | 3.66 | 6.00 | 0.12 | 2.07 | 0.15 |

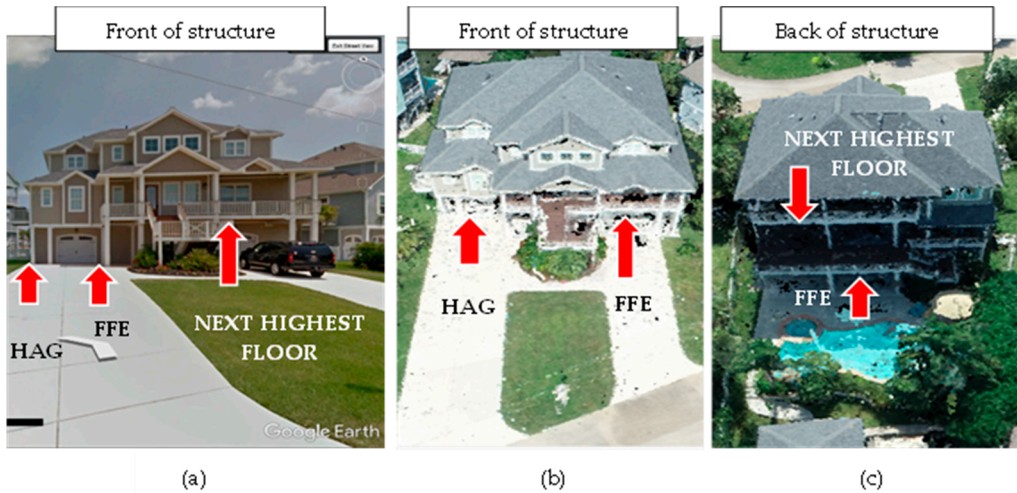

**Figure 5.** Example property 13522 Moyenne Pl demonstrating (**a**) EC measure interpretations seen through google street view (GSV); (**b**) UAS-based measure interpretations displayed on screenshot of generated DPC (front of structure); (**c**) UAS-based measure interpretation displayed on screenshot of generated DPC (back of structure).

UAS based measures on the example property 13,522 Moyenne Pl located in Laffite's was based on utilizing exterior SfM capture, not an in-person evaluation. As seen in Evia, all of the EC FFE values resulted in more reasonable errors when compared to UAS HAG values. Similarly, EC next highest floor measures resulted in more reasonable errors when compared to that of the UAS FFEs. After the interpretation inconsistencies were fully understood, outlying errors between EC and UAS values were paired with the 'correct' corresponding measure taken by both data sources. Again, it is important to understand that these comparative results are not a reflection of how accurate the UAS is in establishing accurate models, rather a comparison of the UAS based estimates to that of traditional EC measures recorded by different surveyors across multiple private companies.

In addition to these challenges, the RTK-UAS coupled with the photogrammetry approach did present limitations with respect to data acquisition and field labor workflows. Firstly, the UAS for SfM capture requires optimal over-head sun lighting, wind conditions less than 25 mph, and zero rain events. These conditions were not only important for data acquisition but also UAS operator safety under the Federal Aviation Administration (FAA) [41]. While traditional and other aerial surveying methods are also limited to these natural conditions, it is important to note that these technologies are weather sensitive. Weather conditions hardly interrupted field labor workflows if the necessary preparation and planning was performed. Secondly, the use of a UAS does require FAA certification and compliance [42]. This meant the completion of a Part 107 UAS flight test was necessary as well as coordination with airports and other geofenced areas if the desired survey area was in a controlled airspace (i.e., Evia). This study also complied with local (City of Galveston) and state (Texas) drone and privacy regulations [43–45]. Although these regulations did not limit data acquisition for the sites selected in this study, other sites of interest may present limitations and operator awareness is an important component for successful UAS flight operations. Thirdly, UAS SfM capture allows for proper modeling of the exterior of static objects and operating within a coastal margin environment revealing both advantages and disadvantages. While the UAS approach excels in capturing flood-sensitive measures in the coastal margin which exhibits foundations of primarily raised and pier types, it does limit FFE capture for inland parcels that potentially contain out-of-sight basements below ground level. Moving objects, such as water and cars were also present in the coastal margin. These phenomena if moving at a high rate would not process properly and had the potential to interfere with other data points if located directly adjacent to a housing structure. Dense vegetation, such as trees and shrubs, occasionally covered a structure and it was not visible from the drone imagery perspective during data derivation. This

limitation conceptualized flight protocol troubleshooting regarding flight altitude and gimbal tilt changes to allow for the capture of more oblique images. Low altitude, high oblique imagery overcame this limitation and future studies should explore combining this approach with terrestrial surveying techniques. Of the 966 total parcels captured, 14 (1.5%) of them were covered by vegetation which did not allow for derivation of measures resulting in the 952 derived parcels. This limitation varied by site selection, but in general, increased vegetation adjacent to structures decreased the likelihood the UAS could capture flood-sensitive measures sought in this study. Lastly, white surfaces, particularly roofs, had the potential to create too great of reflectance depending on the position of the UAS camera lens in relation to the sun. This would result in floating or inconsistently generated points within the DPC. Shadows were processed as lower relief elevations, or in some cases, resulted in blank points within the DPC.

The limitations explained above were primarily a function of UAS flight settings and unavoidable natural conditions. Many of these limitations rarely existed in sites selected, however, if presented were then overcame through UAS troubleshooting and field planning. This study did not consider computational specifications a major limitation but does recognize differences in computer specifications could change processing times and visuals regarding DPC generation. This study did not aim to test differences in computational performances regarding the photogrammetry software, therefore, computer specifications for this study were able to process 'default' or 'recommended' setting workflows within Pix4D. Future studies could explore cloud processing workflows with the potential to reduce processing times. Finally, lack of available data, specifically, EC availability in accordance with residential communities sought in this study was a research limitation. Additionally, the lack of previous studies using this new technology, particularly in dense, built environments, made the methodological approach of this study unique and challenging.

Overall, our study demonstrates the use of a UAS for SfM imagery and photogrammetry to ultimately derive FFEs, and other flood-sensitive data, is viable in achieving measurements accurate to traditional EC methods. Initial comparison results highlighted challenges related to measure interpretation inconsistencies between the UAS based approach and EC surveyors. This study did not aim to critique the NFIP lowest floor guidelines as well as the interpretations associated with them, rather it aimed to compare the actual corresponding measures provided by each source of data. Once this was performed, errors less than previous studies using TST validation were achieved and there was not a statistical difference between UAS based and EC FFE values. UAS technology presents an alternative method to collect large amounts of FFE data regardless of EC challenges discussed and flood insurance policy participation in a cost-effective, efficient manner. UAS derived data is particularly cost-effective with regards to initial equipment requirements and time and labor costs compared to that of traditional EC methods. Not only is this method efficient but it was also successful in capturing a variety of foundation types within a densely built, coastal margin. Future studies should explore an in-depth cost analysis using methods used in this study to determine the cost at the household level. Furthermore, the integration of deep learning and feature fusion with workflows outlined in this study automates the derivation process thus making the UAS based approach even more efficient. Specifically, object and point recognition within a photo or model to determine sought elevational measures.

## 5. Conclusions

In this study, we addressed the lack of alternatives in collecting FFEs and other flood-sensitive data found in ECs by establishing an UAS-based data approach. ECs are obtained through private contractors and serve flood insurers thus making ECs, and the data within limited to administrative COs. Our approach was able to capture all the data found in ECs for four residential communities, 952 parcels, located in Galveston in under 18 working business days. Comparative analysis results of our measures using the UAS and measures within ECs given by the City of Galveston showed the UAS based approach is viable in achieving accurate EC measures. FFEs in particular achieved 0.16 m MAE and were not

statistically different from the EC measures. Preliminary flight protocol testing should be performed for each unique study, however, flight parameters and methods outlined in this study also provide guidance for future studies in achieving similar accuracies within the built environment without the need for excessive GCP establishment. This approach excels in the coastal margin and is adaptable to overcome potential limitations presented in the field within other sites selected. The UAS used in this study was limited to exterior SfM image capture of objects and created derivation challenges for households with dense vegetation. This limitation, as well as the obtainment of basement elevation measures, could be solved by combing terrestrial or in-person surveying techniques. Overall, the use of the UAS captures small-scale spatial and elevational data crucial to other practices and mitigation strategies that regional data sets do not accurately assess.

FFE classification maps can be created and used to identify which exact parcels can expect inundation over others during a flooding event of a given magnitude. This work serves coastal and hazard management teams as well as decision-makers with an alternative to monitoring residential communities subject to high-risk flooding from rain events or storm surges. These areas are subject to increasing change both anthropogenically and environmentally, calling for development modifications due to safety and risk exposures.

Since Hurricane Harvey (2017), The United States Army Corps of Engineers (USACE) Galveston District, The Texas General Land Office (GLO), and various other partners, are working to conceptualize projects that mitigate against future natural disasters and flooding in Galveston and the Greater Houston Area [46,47]. Creating a census of FFEs is a key metric in identifying and prioritizing structures and broader geographies to allocate mitigation funds. The UAS approach described and tested here is a scalable and adaptable method that can be used to better identify and understand structural flood risk.

**Author Contributions:** Conceptualization, W.E.H.; Data curation, N.D.D.; Formal analysis, N.D.D.; Funding acquisition, W.E.H. and B.R.F.; Investigation, N.D.D.; Methodology, N.D.D.; Project administration, W.E.H., S.D.B. and B.R.F.; Resources, W.E.H., S.D.B. and B.R.F.; Software, N.D.D.; Supervision, W.E.H., S.D.B. and B.R.F.; Validation, N.D.D. and W.E.H.; Visualization, W.E.H. and B.R.F.; Writing—original draft, N.D.D.; Writing—review & editing, W.E.H., S.D.B. and B.R.F. All authors have read and agreed to the published version of the manuscript.

**Funding:** This research supported by the Texas Comprehensive Research Fund and Center for Texas Beaches and Shores at Texas A&M University Galveston.

**Data Availability Statement:** The provided links connect to the Texas Digital Library Texas A&M University Depository containing replication data for the creation of the 3D models used in this study. Specifically, the aerial imagery and ground control points (GCPs) for the residential communities Silk Stocking District, Evia, Campeche Cove, and Lafitte's Cove located on Galveston Island, Texas. Texas Digital Library Dataverse: High resolution UAS photogrammetric imagery of residential communities located in Galveston, TX. Available online: https://dataverse.tdl.org/dataverse/diazffe2021 (accessed on 16 February 2022).

**Acknowledgments:** Coastal Resource Manager, Dustin Henry, and Planning Technician, Karen White, with the City of Galveston Planning and Development Division provided Certifications of Occupancy (COs) and Elevation Certificates (ECs) to compare with FFEs obtained using the UAS. CO and EC data is limited due to voluntary participation in the National Flood Insurance Policy (NFIP). 70 ECs were cross-referenced with housing units located in the communities used for this research. Benjamin M. Ritt contributed to sharing another Phantom 4 RTK unit, rod, and remote controller (RC), to link with owned Phantom 4 RTK unit, rod, and RC, ultimately to established known positions and ground control positions (GCPs). Known elevational benchmarks provided by either the National Ocean and Atmospheric Administration (NOAA) or various other private companies and organizations, were not available in certain residential communities.

**Conflicts of Interest:** The authors declare no conflict of interest. The funders had no role in the design of the study; in the collection, analyses, or interpretation of data; in the writing of the manuscript, or in the decision to publish the results.

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
