# Peer review of "Deriving First Floor Elevations within Residential Communities Located in Galveston Using UAS Based Data"

_drones, doi:10.3390/drones6040081_

Round 1

Reviewer 1 Report

The reviewed article was submitted to the journal Drones. The authors of the article attempt to analyze the UAS system. The article definitely meets the requirements of a scientific text. Formally it is flawless. However, analyzing its content, I come to the conclusion that even in such a short form as the article, there would be enough space to develop any of the problematic issues so clearly outlined in the conclusions.  Despite the correct structure in the article, several corrections or clarifications should be added, namely:
1. Add and analyze more literature regarding the subject matter discussed.
2. More detailed discussion of terrain analysis methods and what algorithms are used for this purpose.
3. In the conclusion give the advantages and disadvantages of the discussed system.

Author Response

Please see the attachment. Thank you for your comments. 

Reviewer 2 Report

This paper is well-written, and the technical content is sound and interesting. I have some comments to be discussed with the authors. Therefore, this paper should be re-reviewed before acceptance.
1.    Starting from line 226, key findings are listed. Quite some settings are recommended here. However, it is not clear how the others performed. 
2.    This paper focuses on the utilization of UAS. In my opinion, the discussion on the operational aspects is not sufficient, such as risk management, potential bureaucratic procedures, and rights and privacy, which are very important components of UAS missions.
3.    It is understandable to simply show what is the right use of UAS for the study. What about the unsuccessful experience you learned from the experiments that might also be helpful?
4.    Nit-picking, consider adjusting the font size and the layout of table 6 for better readability.

Author Response

(The authors gave the same response as above.)

Round 2

Reviewer 2 Report

The authors have addressed my concerns. Thank you!